# Histoplasmosis and Tuberculosis Co-Occurrence in People with Advanced HIV

**DOI:** 10.3390/jof5030073

**Published:** 2019-08-09

**Authors:** Diego H. Caceres, Audrey Valdes

**Affiliations:** 1Mycotic Diseases Branch, Centers for Disease Control and Prevention, Atlanta, GA 30333, USA; 2Studies in Translational Microbiology and Emerging Diseases (MICROS) Research Group, School of Medicine and Health Sciences, Universidad del Rosario, 11011 Bogota, Colombia; 3Service de Maladies Infectieuses et Tropicales, Centre Hospitalier de Cayenne, 97306 Cayenne, French Guiana

**Keywords:** histoplasmosis, tuberculosis, HIV, AIDS, co-occurrence, epidemiology, diagnosis, treatment

## Abstract

Distinguishing between histoplasmosis, tuberculosis (TB), and co-occurrence of disease is a frequent dilemma for clinical staff treating people with advanced Human Immunodeficiency Virus (HIV) infection. This problem is most frequently observed in clinical settings in countries where both diseases are endemic. It is also a challenge outside these endemic countries in HIV clinics that take care of patients coming from countries with endemic histoplasmosis and TB. The gold standard for diagnosis of both of these diseases is based on conventional laboratory tests (culture, histopathology and special stains). These tests have several limitations, such as lack of laboratory infrastructure for handling isolates (biosafety level 3), shortage of laboratory staff who have appropriate training and experience, variable analytical performance of tests and long turn-around time. Recently, novel rapid assays for the diagnosis of histoplasmosis and TB became available. However, this technology is not yet widely used. Mortality in immunocompromised patients, such as people with advanced HIV, is directly linked with the ability to rapidly diagnose opportunistic diseases. The aim of this review is to synthesize the main aspects of epidemiology, clinical characteristics, diagnosis and treatment of histoplasmosis/TB co-occurrence in people with advanced HIV.

## 1. Introduction

A wide variety of opportunistic diseases occurs in people with advanced HIV depending on CD4 counts. The lower this count, the higher the risk of multiple infections. Tuberculosis (TB) is one of the main opportunistic infections around the world, with an estimated burden of 1.3 million new disease cases/year, and 300,000 deaths/year among people living with HIV (PLHIV) [1]. Histoplasmosis, caused by the dimorphic fungus *Histoplasma capsulatum*, has been reported worldwide, but most cases have been identified in the Americas [2]. In contrast to TB, the burden of histoplasmosis is less well known. Studies from the 1950s using intradermal reaction (IDR) with histoplasmin revealed the high prevalence of people with exposure to *Histoplasma* in the general population in the Americas and some regions in Asia and Africa [3,4,5]. One epi-serological study done in 1960 in Michigan, United States, showed that 26% of the general population had intradermal reaction against tuberculin or histoplasmin, and 5% to both antigens [6]. 

No global estimate of the histoplasmosis burden in PLHIV is available, and in many endemic countries, estimating histoplasmosis burden is not feasible due to lack of data [7,8,9,10]. A recent report provided an estimate of the histoplasmosis burden in PLHIV in Latin American countries using data from previous IDR studies [11]. Authors estimated that in some countries, the prevalence of histoplasmosis and TB were equivalent, but histoplasmosis caused more deaths compared with TB. Studies based on autopsies in Brazil and Peru identified 7%–28% of deaths as related to tuberculosis and 12%–21% of deaths related to histoplasmosis [12,13]. A retrospective review of histoplasmosis cases reported mortality rates of 20%–53% [14]. 

The poor understanding of histoplasmosis prevalence and the lack of disease awareness among clinicians are directly linked to the clinical confusion between histoplasmosis and tuberculosis. Patients with histoplasmosis are often initially diagnosed and treated for TB, resulting in treatment failure [15,16]. 

## 2. Materials and Methods 

Literature search: We searched the following databases for the terms histoplasmosis, HIV and tuberculosis, including synonyms in the title, abstract, keywords or subject headings: Medline (Ovid), Embase (Ovid), CAB Abstracts (Ovid), Global Health (Ovid), Scopus, the Cochrane Library, PubMed Central and LILACS. These searches were limited to studies published in English, Spanish and Portuguese. We searched for reports of cohorts of patients with progressive disseminated histoplasmosis and HIV, and frequency of TB disease co-occurrence. We aimed to synthesize the main aspects of epidemiology, clinical characteristics, diagnosis and treatment of histoplasmosis/TB co-occurrence in people with advanced HIV.

## 3. Epidemiology of Histoplasmosis/TB Co-Occurrence in People with Advanced HIV

Symptoms of histoplasmosis are nonspecific and often indistinguishable from those of other infectious diseases, especially TB. This complicates the clinical diagnosis and treatment. Co-occurrence of TB has been reported in several cohorts of patients with histoplasmosis and advanced HIV from Latin American countries (Figure 1). Patients with this co-occurrence share the following common characteristics: (1) They are often highly immunosuppressed (median CD4 T cell: 30 cells/mm^3^). (2) In most of these studies, TB diagnosis was done by direct microscopic observation of acid-fast-bacilli (AFB) in specimens from symptomatic people, without microbiological confirmation by culture. (3) Treatment failures were frequently reported [17,18,19,20,21,22,23,24,25,26]. Based on these reports, it is important to highlight the key role of laboratory testing in the diagnosis of histoplasmosis or TB, or the co-occurrence of these two conditions. In order to a quickly diagnose and treat patients, and improve patient outcomes, accurate laboratory testing should be done if there is any suspicion that a patient might have histoplasmosis, TB or other diseases.

## 4. Clinical Characteristics

Histoplasmosis and tuberculosis are frequently identified in patients with advanced HIV (CD4 T cells under 200/mm^3^). Two clinical presentations are frequently described, the pulmonary and the progressive disseminated form. However, it is often difficult to differentiate between the two diseases [27,28,29]. Common symptoms include fever, weight loss and respiratory symptoms. Chest imaging often shows localized pulmonary lesions or miliary nodules. Multiple organs can become involved including: lymph nodes, bone marrow, spleen, liver, digestive tract, adrenal gland and central nervous system [30,31]. In clinical descriptions of people with histoplasmosis and TB co-occurrence, fever was the most common manifestation (75%), and half (50%) of the cases presented lymph node enlargement, gastrointestinal (mostly abdominal) pain and respiratory symptoms, dominated by an isolated cough. Three of seven patients (43%) with a reported outcome died. Histoplasmosis diagnosis was delayed in non-endemic countries [32,33,34,35,36,37].

Several studies have tried to identify clinical patterns for differentiating histoplasmosis and TB disease in PLHIV. A study from French Guiana used a multivariate model to compare symptoms in HIV patients infected with either histoplasmosis or TB alone and identified a specific clinical and biological profile. Within this profile, disseminated histoplasmosis appeared to have a prominent gastrointestinal tract involvement, while disseminated tuberculosis had a concomitant pulmonary expression [38]. In a study from Colombia, a multidimensional statistical method was used to identify a clinical profile in patients with histoplasmosis and advanced HIV; results on this study were similar to the results reported from French Guiana [20]. However, it was not possible to unambiguously diagnose histoplasmosis/TB co-occurrence in PLHIV based solely on the clinical evaluation of signs and symptoms. This finding highlights the need for specific diagnostic tests for histoplasmosis, TB and co-occurrence, in order to reduce diagnostic delays and improve patient outcomes. 

Gold standards for the diagnosis of histoplasmosis and TB are based on conventional laboratory methods that include histopathological analysis, special stains and culture [2,39,40,41,42]. For the diagnosis of pulmonary TB, analysis of sputum samples by microscopic observation of AFB is widely available and can provide results quickly and with acceptable analytical performance. On the other hand, for the diagnosis of pulmonary histoplasmosis, sputum samples are not appropriate. In order to increase the sensitivity, it is necessary to analyze invasive respiratory samples like bronchoalveolar lavage (BAL) fluid, or other lower respiratory tract specimens [2,27,41]. For the diagnosis of disseminated TB and histoplasmosis, it is necessary to test samples from the tissue involved; yet obtaining such invasive samples could present a risk for complications in critically ill patients. Blood cultures and bone marrow culture are useful to diagnose disseminated TB and histoplasmosis. High sensitivity is observed when samples are processed using tests based on lysis systems (>80% sensitivity) [43,44,45,46]. Conventional laboratory methods have many challenges, including the need for complex laboratory infrastructure (biosafety level 3 laboratories), personnel with extensive training and competency in microbiology, delays of several weeks for final laboratory results and variable analytical performance. For example, the sensitivity of histoplasmosis culture testing ranges from 42% to 74% and depends on the clinical form of the disease and specimen quality [2,27,41].

For histoplasmosis diagnosis, antibody detection tests are less sensitive in PLHIV, with a sensitivity ranging between 38% to 70% [27,41,47]. Assays for intradermal reaction or interferon-γ release assays are not commercially available. Recently the development of an interferon gamma release assay specific for *Histoplasma capsulatum*, aimed to detect asymptomatic infected individuals was reported; future validations are necessary to know the analytical performance of this assay [48]. Conversely, detection of circulating *Histoplasma* antigen has proven highly sensitive (>90%) [49]. Since 1986, this test has been offered by a private laboratory in the United States of America (MiraVista Diagnostics™, Indianapolis, IN, USA), and some commercial ELISA kits are available in the market. One commercial kit (*Histoplasma* Galactomannan EIA, IMMY™, Norman, OK, USA) has shown high analytical performance with internal and external laboratory reproducibility in a multicenter validation in two Latin American countries [50]. A lateral flow device was recently developed. This technology shows the same analytical performance as the ELISA methodology, but in contrast to the ELISA test, it can be performed in less complex laboratories, requires less time and can potentially reduce the costs associated with the test’s performance [51]. Additionally, antigen detection has increased detection of histoplasmosis cases and reduced mortality in Colombia, India, Guatemala and Brazil [19,22,52,53]. 

Diagnosis of infection with *Mycobacterium tuberculosis* can be performed by antibody and antigen detection [40]. Tuberculin skin testing and interferon-γ release assays are available for diagnosis of latent *M. tuberculosis* infection. These tests are recommended for people at risk of developing tuberculosis. Those with a positive test result can be treated to reduce the possibility of developing active TB disease [40,54]. A lateral flow urine lipoarabinomannan assay (LF-LAM) is also available (Alere Determine™ TB LAM, Abbott, Chicago, IL, USA). The World Health Organization (WHO) recommended the LF-LAM to assist in the diagnosis of TB in PLHIV with signs and symptoms of TB and low CD4 cell count (<100 cells/μL) [55]. 

There are no commercial PCR kits for the diagnosis of histoplasmosis directly on patient specimens. The only available commercial kit is for the identification of culture isolates. Several in-house PCR assays have been developed, but the analytical performance of these tests is variable [56,57,58,59,60,61,62]. This variation could be due to the lack of consensus on gene targets, DNA extraction methods, laboratory procedures, sample type, etc. Molecular tests look promising for histoplasmosis diagnosis, but further investigations are needed for protocol standardization.

Many different alternatives for TB diagnosis and antimicrobial resistance are available commercially. The Xpert MTB/RIF Ultra test (Cepheid, Sunnyvale, CA, USA) is one of the most available tests. This is a non-culture based real-time PCR, able to detect *M. tuberculosis* DNA and mutations associated with resistance to rifampicin direct on a patient’s specimens in less than two hours [63,64]. Other molecular commercial tests are available, including RealTime MTB (Abbott, IL, USA), FluoroType MTBDR (Bruker, Billerica, MA, USA), Anyplex™ II MTB/XDR (Seegene, Seoul, Korea) and BD MAX MDR-TB (Beckton, Dickinson and Company, Franklin Lakes, NJ, USA). Currently, Matrix-assisted laser desorption ionization-time-of-flight (MALDI-TOF) mass spectrometry is one of the best options for identification of *Mycobacterium* species isolates. This technology is quick and highly accurate for isolate identification.

In November 2018, the World Health Organization (WHO) published the first essential diagnostics list [65]. This list contains 113 products, 58 for detection and diagnosis of a wide range of common conditions, and the remaining 55 for diagnosis and monitoring of priority diseases such as HIV, tuberculosis, malaria, hepatitis B and C, human papillomavirus and syphilis. This first version did not include any laboratory test for the diagnosis of histoplasmosis, but the Second WHO Model List of Essential In Vitro Diagnostics published at the end of July 2019, includes the *Histoplasma* antigen testing [66]. 

## 5. Treatment of Histoplasmosis/TB Co-Occurrence

In 2017, the WHO released guidelines to address advanced HIV disease, including screening, diagnosis and treatment of the most common opportunistic diseases in HIV-infected people [67]. These guidelines noted that consideration should be given to regional differences in comorbidities and coinfections but did not specifically provide guidance on diagnosis and management of histoplasmosis. The Infectious Diseases Society of America (IDSA) developed histoplasmosis treatment guidelines in 2007. These guidelines have limitations for implementation outside of the United States, as they do not follow the guideline development methodology that has been standardized by the WHO and do not describe management of TB co-occurrence in PLHIV with histoplasmosis [68].

The IDSA guidelines recommend treatment according to disease severity: moderate to severe, and mild to moderate. For moderately severe to severe disease, the recommended treatment includes liposomal amphotericin B (3.0 mg/kg daily, 1–2 weeks), followed by oral itraconazole (200 mg 3 times daily for 3 days and then 200 mg twice daily for a total of at least 12 months). Deoxycholate amphotericin B (0.7–1.0 mg/kg daily) is an alternative in patients who are at a low risk for nephrotoxicity. For mild to moderate disease, treatment includes itraconazole (200 mg 3 times daily for 3 days and then twice daily for at least 12 months). Blood levels of itraconazole and *Histoplasma* antigen concentrations (sera or urine) should be measured during therapy. *Histoplasma* antigen concentrations should be monitored for 12 months after the end of the therapy to detect early signs of relapse [68]. Table 1 summarizes the recommendations for treatment of histoplasmosis and itraconazole interactions.

Itraconazole can interactwith several antibacterial and retroviral (ART) medications. The TB treatment medications rifampicin and rifabutin can decrease levels of itraconazole, and itraconazole can increase levels of rifabutin. On the other hand, the ART drugs, efavirenz (EFV) and nevirapine (NVP) reduces itraconazole blood levels, and lopinavir/ritonavir (LPV/r) and atazanavir/ritonavir (ATV/r) increases itraconazole blood levels. For that reason, it is necessary to monitor itraconazole blood concentration during time of treatment. Currently, there are no guidelines or recommendations for the treatment of the histoplasmosis/TB co-occurrence. A study from Colombia describes a case series of 12 PLHIV with histoplasmosis/TB co-occurrence treated with itraconazole. The authors describe two groups of treatment: (1) itraconazole plus rifampicin + isoniazid + pyrazinamide + ethambutol (RHZE) (*n* = 6 patients), and (2) itraconazole plus isoniazid + pyrazinamide + ethambutol (HZE), plus a quinolone (*n* = 6 patients). In group 1, they found therapeutic itraconazole blood levels in 0 of 3 patients measured, treatment success in 4 of 6 patients, 1 patient’s death and 1 histoplasmosis relapse. In group 2, they found therapeutic itraconazole blood levels in 2 of 2 patients measured, success treatment in 5 of 6 patients, and one patient was lost to follow-up [69]. A problem in this patient population is that the ARV introduction is usually delayed by two weeks due to the risk of immune reconstitution inflammatory syndrome [70]. 

## 6. Conclusions

People living with HIV are at a high risk for developing multiple opportunistic diseases, and the evaluation of clinical signs and symptoms is not enough to establish a final diagnosis. Although recent technological advances have improved diagnostic accuracy of advanced HIV-related opportunistic diseases, these technologies are not available in many regions around the world. In addition, fungal opportunistic diseases are neglected in most of the HIV programs worldwide, resulting in inadequate health system planning and missed opportunities to save lives. Focus areas are the role of education, networking and access to accurate laboratory tests used to promptly confirm correct diagnoses and provide appropriate therapies.

Based on the findings of this review, tuberculosis co-occurrence is a frequent problem in people with histoplasmosis and advanced HIV; most of the reports came from the Latin American region and involved very ill patients (median CD4 T cell: 30 cells/mm^3^). The lack of epidemiological data, diseases awareness, rapid accurate diagnostics tests and treatment guidelines are the main threats to this patient population. These gaps could be improved by: (1) improving the histoplasmosis epidemiological data through the implementation of surveillance systems; (2) increasing histoplasmosis disease awareness by implementing educational strategies, such as the inclusion of histoplasmosis in national HIV program guidelines; and (3) increasing awareness of novel rapid diagnostic tests for histoplasmosis, TB and other opportunistic diseases.

## Figures and Tables

**Figure 1 jof-05-00073-f001:**
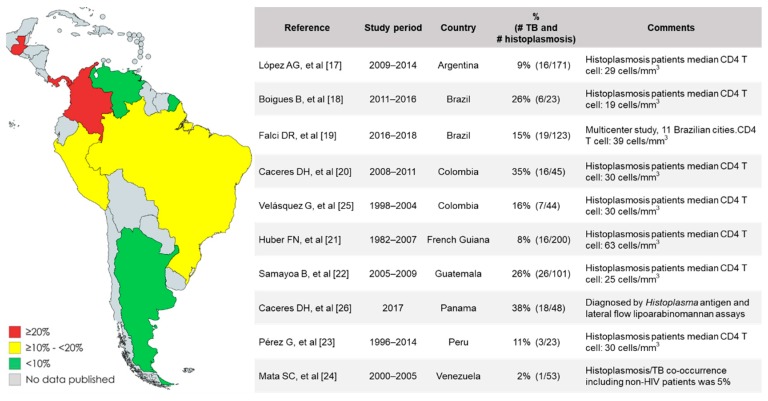
Reports of cohorts of people with histoplasmosis and advanced HIV: frequency of tuberculosis (TB) co-occurrence.

**Table 1 jof-05-00073-t001:** Recommendation for treatment of histoplasmosis in persons living with HIV (PLHIV).

**Itraconazole drug interactions**	**Antibacterial for TB treatment**: • Rifampicin and rifabutin may decrease itraconazole blood levels • Itraconazole may increase blood levels of rifabutin **Antiretroviral:** • Efavirenz (EFV) and nevirapine (NVP) reduce itraconazole blood levels • Lopinavir/ritonavir (LPV/r) and atazanavir/ritonavir (ATV/r) increase itraconazole blood levels Monitoring itraconazole blood levels.
**Recommendations for treatment of progressive disseminated histoplasmosis** (Infectious Diseases Society of America (71) guidelines)	**Moderately severe to severe disease**: • Liposomal amphotericin B (3.0 mg/kg daily, 1–2 weeks), followed by oral itraconazole (200 mg 3 times daily for 3 days and then 200 mg twice daily for a total of at least 12 months in disseminated form) [68]. **For mild to moderate disease**: • Itraconazole (200 mg 3 times daily for 3 days and then twice daily for at least 12 months) [68]. Blood levels of itraconazole and *Histoplasma* antigen concentrations (sera or urine) should be measured during therapy. Monitor *Histoplasma* antigen concentrations for 12 months after end of therapy with the aim of early identification of histoplasmosis relapse [68].
**Treatment of histoplasmosis/TB** **co-occurrence**	At the time of this review, there were no guidelines for the treatment of histoplasmosis/TB co-occurrence. A report from Colombia used fluoroquinolone as an alternative agent in place of rifampicin for tuberculosis [69]. This was a descriptive study with evidence of a very low certainty.

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
