# Peer review of "Histoplasmosis and Tuberculosis Co-Occurrence in People with Advanced HIV"

_jof, 2019, doi:10.3390/jof5030073_

Round 1

Reviewer 1 Report

This is an interesting review, especially in the context of advanced HIV disease. 

A few comments to improve the manuscript are here below: 

Line 23-  Correct the grammar in the sentence

Line 42 - I thought it would be appropriate to add these references: 

PLoS Negl Trop Dis. 2018 Jan 18;12(1):e0006046. doi: 10.1371/journal.pntd.0006046. eCollection 2018 Jan.

Med Mycol. 2016 Mar;54(3):295-300. doi: 10.1093/mmy/myv081. Epub 2015 Nov 2.

Line 68- have you considered suspected histoplasmosis in post TB lung disease? other than among patients with TB treatment failure?

Line 70- correct the grammar

For figure 1, did you not find published case series from other regions?

Line 93 - replace digestive with gastro intestinal tract

Line 113-  personnel, not personal

Line 127 where the different tests are discussed - Include costs of diagnostics here

Line 145- Would be more specific and mention the Xpert ultra, which appears to be more sensitive

Line 207 - If no other published cases of co-infection are available from other regions, perhaps an addition to the conclusion would be that this is more prevalent in Latin America.

Author Response

This is an interesting review, especially in the context of advanced HIV disease.

A few comments to improve the manuscript are here below:

Line 23- Correct the grammar in the sentence

Corrected. Thanks

Line 42 - I thought it would be appropriate to add these references: PLoS Negl Trop Dis. 2018 Jan 18;12(1):e0006046. doi: 10.1371/journal.pntd.0006046. eCollection 2018 Jan. Med Mycol. 2016 Mar;54(3):295-300. doi: 10.1093/mmy/myv081. Epub 2015 Nov 2.

We added the references in the introduction.

Line 68- have you considered suspected histoplasmosis in post TB lung disease? other than among patients with TB treatment failure?

No, we focus the literature search on cohort of people with HIV and histoplasmosis, them, we extract data of final diagnose of TB on those cohorts

Line 70- correct the grammar

Corrected. Thanks

For figure 1, did you not find published case series from other regions?

not cases series found following criteria described in the material and methods section. There are some cases repots mostly from Asia.

Line 93 - replace digestive with gastro intestinal tract

replaced.

Line 113- personnel, not personal

replaced.

Line 127 where the different tests are discussed - Include costs of diagnostics here

I have restrictions from CDC to discuss cots

Line 145- Would be more specific and mention the Xpert ultra, which appears to be more sensitive

Added

Line 207 - If no other published cases of co-infection are available from other regions, perhaps an addition to the conclusion would be that this is more prevalent in Latin America.

Thanks, we modified discussion as follows: Based on finding of this review, tuberculosis co-occurrence is a frequent problem in persons with histoplasmosis and advanced HIV; most of the reports coming from the Latin America Region, and involving very ill patients (median CD4 T cell: 30 cells/mm3).”

Reviewer 2 Report

The authors, in this review, describe the co-occurrence of TB and Histoplasmosis in advanced HIV+ve patients. The review article is important and novel and highlights the co-occurrence of the diseases that is often overlooked. However, authors need to address the followings.

Title may be changed to include specific geographical area such as "in Latin America" May need discussion on if the CD4 T cells to be below 100 (or very low) to see co-infections.  Are there any predisposing factors for the co-infection other than CD4 T cells? Is delayed ARV recommended to avoid IRIS?

Author Response

The authors, in this review, describe the co-occurrence of TB and Histoplasmosis in advanced HIV+ve patients. The review article is important and novel and highlights the co-occurrence of the diseases that is often overlooked. However, authors need to address the followings.

Title may be changed to include specific geographical area such as "in Latin America" May need discussion on if the CD4 T cells to be below 100 (or very low) to see co-infections. Are there any predisposing factors for the co-infection other than CD4 T cells? Is delayed ARV recommended to avoid IRIS?

Thanks for your comments. We preferred to keep the title and do not include a geographical restriction. We modified discussion as follows: Based on finding of this review, tuberculosis co-occurrence is a frequent problem in persons with histoplasmosis and advanced HIV; most of the reports coming from the Latin America Region, and involving very ill patients (median CD4 T cell: 30 cells/mm3).”

We cannot recommended delayed ARV due the lack of specific histoplasmosis guidelines.

Reviewer 3 Report

This is an interesting and timely review; there are many points of congruence between TB and histoplasmosis, which have been brought together in this review in the setting of HIV. I was reminded of the fact that, in the early days of the description of histoplasma disease (late 50s), as many as 6% of patients in US TB sanatoria were found to be infected with H. capsulatum (Walls et al. J Lab Clin Med, 1958, 51:266).

Overall, this is a nicely presented review, although I'd urge the authors to undertake a close language check. (For example, line 113 has 'personal' instead of intended word 'personnel'.)

However, my major concern with this review is that although the title is sufficiently generalized, the presentation of the data from the target group (PLHIV) seems to be geographically restricted to largely Central and South America.

This is a relevant region, without a doubt. Studies by Adenis et al. (two of which have been cited in this manuscript) show that histoplasmosis has overtaken TB as the leading cause of death in PLHIV in this particular region.

But given the importance of this discussion, I do think that the scope of the information presented should be expanded to include North America as well, especially since, far from the early days of histoplasmin skin tests, histoplasmosis is reasonably well-described in the US now (including from the CDC) -- with current data even indicating an expansion of the presumed endemic zone. I hope that the authors would give this due consideration.

If the US data is considered out of the scope of this manuscript, I'd recommend restricting the title statement accordingly.

Another minor matter. Given the importance of IGRAs in diagnosis of LTBI, perhaps similar nascent efforts to diagnose latent histoplasmosis deserves a general mention (even though current studies have not specifically looked at the PLHIV cohort). One such study from the same part of the world (that I know of) is a recent paper by Rubio-Carrasquilla et al. (DOI: 10.1093/mmy/myy131). 

Author Response

This is an interesting and timely review; there are many points of congruence between TB and histoplasmosis, which have been brought together in this review in the setting of HIV. I was reminded of the fact that, in the early days of the description of histoplasma disease (late 50s), as many as 6% of patients in US TB sanatoria were found to be infected with H. capsulatum (Walls et al. J Lab Clin Med, 1958, 51:266).

Overall, this is a nicely presented review, although I'd urge the authors to undertake a close language check. (For example, line 113 has 'personal' instead of intended word 'personnel'.)

However, my major concern with this review is that although the title is sufficiently generalized, the presentation of the data from the target group (PLHIV) seems to be geographically restricted to largely Central and South America.

This is a relevant region, without a doubt. Studies by Adenis et al. (two of which have been cited in this manuscript) show that histoplasmosis has overtaken TB as the leading cause of death in PLHIV in this particular region.

But given the importance of this discussion, I do think that the scope of the information presented should be expanded to include North America as well, especially since, far from the early days of histoplasmin skin tests, histoplasmosis is reasonably well-described in the US now (including from the CDC) -- with current data even indicating an expansion of the presumed endemic zone. I hope that the authors would give this due consideration.

If the US data is considered out of the scope of this manuscript, I'd recommend restricting the title statement accordingly.

Another minor matter. Given the importance of IGRAs in diagnosis of LTBI, perhaps similar nascent efforts to diagnose latent histoplasmosis deserves a general mention (even though current studies have not specifically looked at the PLHIV cohort). One such study from the same part of the world (that I know of) is a recent paper by Rubio-Carrasquilla et al. (DOI: 10.1093/mmy/myy131).

R. Thanks for your comments. We found some reports from the US, but this is data where from old studies, 80’s and early 90’s. That was the reason why we excluded these reports.

We added Rubio-Carrasquilla et al report “Assays for intradermal reaction or interferon-γ release assays are not commercially available. Recently was reported the development of an interferon gamma release assay specific for Histoplasma capsulatum, aim to detect asymptomatic infected individuals; future validations are necessary to know the analytical performance of this assay.”